# Transposase-assisted tagmentation of RNA/DNA hybrid duplexes

Bo Lu[1,2†], Liting Dong[1,2†], Danyang Yi[1†], Meiling Zhang[1†], Chenxu Zhu[1], Xiaoyu Li[1], Chengqi Yi[1,2,3]*

[1]State Key Laboratory of Protein and Plant Gene Research, School of Life Sciences, Peking University, Beijing, China; [2]Peking-Tsinghua Center for Life Sciences, Peking University, Beijing, China; [3]Department of Chemical Biology and Synthetic and Functional Biomolecules Center, College of Chemistry and Molecular Engineering, Peking University, Beijing, China

**Abstract** Tn5-mediated transposition of double-strand DNA has been widely utilized in various high-throughput sequencing applications. Here, we report that the Tn5 transposase is also capable of direct tagmentation of RNA/DNA hybrids in vitro. As a proof-of-concept application, we utilized this activity to replace the traditional library construction procedure of RNA sequencing, which contains many laborious and time-consuming processes. Results of Transposase-assisted RNA/DNA hybrids Co-tagmEntation (termed 'TRACE-seq') are compared to traditional RNA-seq methods in terms of detected gene number, gene body coverage, gene expression measurement, library complexity, and differential expression analysis. At the meantime, TRACE-seq enables a cost-effective one-tube library construction protocol and hence is more rapid (within 6 hr) and convenient. We expect this tagmentation activity on RNA/DNA hybrids to have broad potentials on RNA biology and chromatin research.

*For correspondence:
chengqi.yi@pku.edu.cn

†These authors contributed equally to this work

Competing interests: The authors declare that no competing interests exist.

## Introduction

Transposases exist in both prokaryotes and eukaryotes and catalyze the movement of defined DNA elements (transposon) to another part of the genome in a 'cut and paste' mechanism (*Kleckner, 1981*; *Finnegan, 1989*; *Curcio and Derbyshire, 2003*). Taking advantage of this catalytic activity, transposases are widely used in many biomedical applications: for instance, an engineered, hyperactive Tn5 transposase from *E. coli* can bind to synthetic 19 bp mosaic end-recognition sequences appended to Illumina sequencing adapters (termed 'Tn5 transposome') (*Adey et al., 2010*) and has been utilized in an in vitro double-stranded DNA (dsDNA) tagmentation reaction (namely simultaneously fragment and tag a target sequence with sequencing adaptors) to achieve rapid and low-input library construction for next-generation sequencing (*Adey et al., 2010*; *Goryshin and Reznikoff, 1998*; *Picelli et al., 2014a*; *Caruccio, 2011*; *Ramsköld et al., 2012*; *Gertz et al., 2012*). In addition, Tn5 was also used for in vivo transposition of native chromatin to profile open chromatin, DNA-binding proteins and nucleosome position ('ATAC-seq') (*Buenrostro et al., 2013*). While Tn5 has been broadly adopted in high-throughput sequencing, bioinformatic analysis and structural studies reveal that it belongs to the retroviral integrase superfamily that act on not only dsDNA but also RNA/DNA hybrids (for instance, RNase H). Despite the distinct substrates, these proteins all share a conserved catalytic RNase H-like domain (*Figure 1a*; *Yang and Steitz, 1995*; *Savilahti et al., 1995*; *Nowotny, 2009*; *Rice and Baker, 2001*). Given their structural and mechanistic similarity, we attempted to ask whether or not Tn5 is able to catalyze co-tagmentation reactions to both the RNA and DNA strands of RNA/DNA hybrids (*Figure 1b*), in addition to its canonical function of dsDNA transposition. In this study, we tested this hypothesis and found that indeed Tn5 possesses in vitro tagmentation activity towards both strands of RNA/DNA hybrids. As a

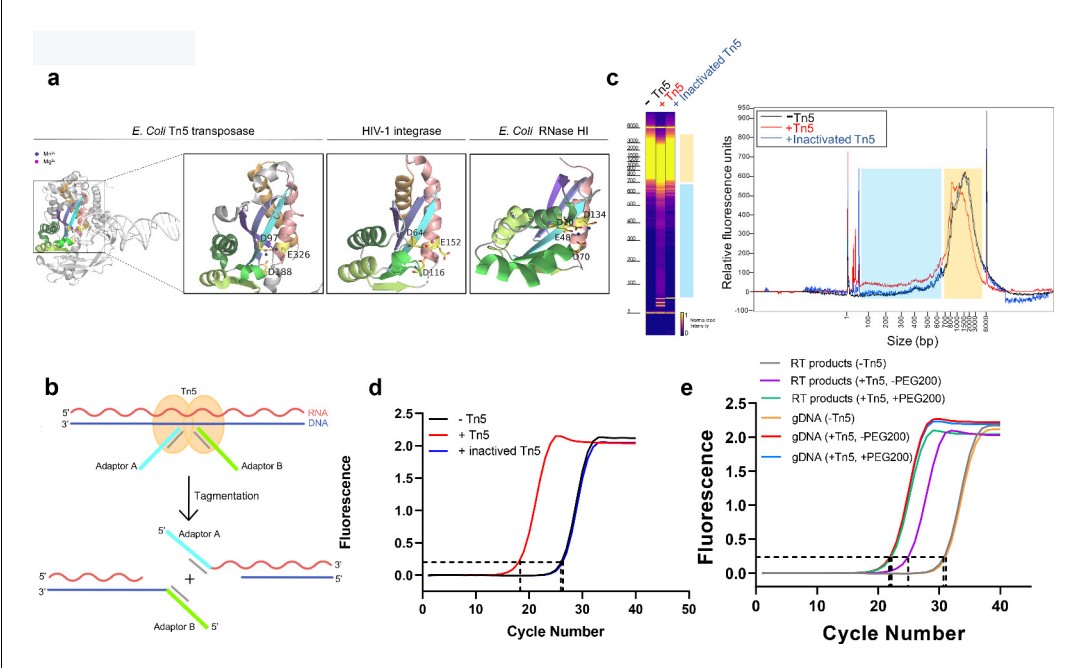

**Figure 1.** Tn5 transposome has direct tagmentation activity on RNA/DNA hybrid duplexes. (a) Crystal structure of a single subunit of *E. coli* Tn5 Transposase (PDB code 1MM8) complexed with ME DNA duplex, and zoom-in views of the conserved catalytic core of Tn5 transposase, HIV-1 integrase (PDB code 1BIU), and *E. coli* RNase HI (PDB code 1G15), all of which are from the retroviral integrase superfamily. Active-site residues are shown as sticks, and the $Mn^{2+}$ and $Mg^{2+}$ ions are shown as deep blue and magenta spheres. (b) Schematic of Tn5-assisted tagmentation of RNA/DNA hybrids. (c) Gel pictures (left) and peak pictures (right) represent size distributions of HEK293T mRNA-derived RNA/DNA hybrid fragments after incubation without Tn5 transposome, with Tn5 transposome, and with inactivated Tn5 transposome. The blue and orange patches denote small and large fragments, respectively. (d) qPCR amplification curve of tagmentation products of HEK293T mRNA-derived RT samples with Tn5 treatment, with inactivated Tn5 treatment, or without Tn5 treatment. Average Ct values of two technical replicates are 18.06, 26.25 and 26.41, respectively. (e) qPCR amplification curve of tagmentation products of HEK293T mRNA-derived RT products samples and gDNA samples under different conditions. (Average Ct values of three technical replicates: RT products sample without Tn5 treatment = 30.38; RT products sample with PEG200 = 21.94; RT products sample without PEG200 = 25.23; gDNA sample without Tn5 treatment = 30.71; gDNA sample with PEG200 = 21.15; gDNA sample without PEG200 = 21.19).

The online version of this article includes the following source data and figure supplement(s) for figure 1:

**Source data 1.** qPCR Ct values of tagmentation products of samples under different conditions in *Figure 1d and e*.

**Figure supplement 1.** Tagmentation activity of Tn5 transposome on RNA/DNA hybrids.

proof of concept, we apply such <u>T</u>ransposase-assisted <u>RNA/DNA</u> hybrids <u>C</u>o-tagm<u>E</u>ntation (TRACE-seq) to achieve rapid and low-cost RNA sequencing starting from total RNA extracted from 10,000 to 100 cells. We find that TRACE-seq performs well when compared with conventional RNA-seq methods in terms of detected gene number, gene expression measurement, library complexity, GC content and differential expression analysis, although TRACE-seq shows bias in gene body coverage and is not strand-specific. At the same time, it avoids many laborious and time-consuming steps in traditional RNA-seq experiments. Such Tn5-assisted tagmentation of RNA/DNA hybrids could have broad applications in RNA biology and chromatin research.

## Results

To test whether Tn5 transposase has tagmentation activity on RNA/DNA hybrids, we prepared RNA/DNA duplexes by performing mRNA reverse transcription. We first validated the efficiency of reverse transcription and the presence of RNA/DNA duplexes using a model mRNA sequence (IRF9,~1000 nt) as template (*Figure 1—figure supplement 1a*). We then subjected the prepared RNA/DNA hybrids from 293T mRNA to Tn5 transposome, heat-inactivated Tn5 transposome and a blank control (without Tn5), respectively (see Methods). The hybrids were then recovered and their length distribution was analyzed by Fragment Analyzer (*Figure 1c*). Comparing with the heat-

inactivated Tn5 sample or the blank control sample, the Tn5 transposome sample exhibited a modest but clear smear signal corresponding to small fragments ranging from ~30–650 base-pair (bp) (the blue patches in *Figure 1c*). Consistent with the fragmentation event, we also observed a down shift of large fragments ranging from ~700–4000 bp (the orange patches in *Figure 1c*). In addition, the fragmentation efficiency increased in a dose-dependent manner with the transposome, suggesting that fragmentation of RNA/DNA hybrids is dependent on Tn5 (*Figure 1—figure supplement 1b*).

We next asked whether RNA/DNA hybrids are tagged by Tn5 and performed quantitative polymerase chain reaction (qPCR) quantification for the three samples. We observed that cycle threshold (Ct) value of the Tn5 transposome sample is about eight cycles smaller than the heat inactivated Tn5 sample or the control sample, indicating approximately 256 times more amplifiable products (*Figure 1d*). We also tested different buffer conditions and found that the performance of Tn5 remained similar, indicating the robustness of the Tn5 tagmentation activity (*Figure 1—figure supplement 1c*). Using Sanger sequencing, we validated that the adaptor sequences are indeed ligated to the insert sequences (*Figure 1—figure supplement 1d*).

To compare Tn5 tagmentation efficiency between RNA/DNA hybrids and dsDNA, we performed tagmentation and qPCR on equal amount of mRNA RT products and genomic DNA (gDNA). Average Ct value of the hybrids samples was about four cycles more than gDNA samples, indicating the efficiency of Tn5 toward hybrids is about 1/16 of that of dsDNA (*Figure 1e*). It is known that natural RNA/DNA hybrids favor A-form conformation. Interestingly, in the presence of PEG200, hybrids were found to favor B-form conformation (*Pramanik et al., 2011*), which we expected to make the hybrids a better substrate of Tn5. Indeed, addition of PEG200 diminishes this difference by greatly improving the Tn5 tagmentation efficiency towards hybrids (*Figure 1e*). This result indicates that the conformation of substrates certainly affects the preference of Tn5. We also ruled out the possibility of gDNA contamination in RT products (*Figure 1—figure supplement 1e*). Taken together, under optimized conditions, Tn5 shows significantly improved efficiency towards RNA/DNA hybrids.

As reverse transcriptase could produce dsDNA from RNA/DNA hybrids, we next designed an experiment by eliminating the RT component and directly assess tagmentation activity using annealed RNA/DNA hybrids where no dsDNA is possible. We annealed in vitro transcribed and purified ssRNA (CLuc, 150 nt, GC% = 51%) with chemically synthesized complementary ssDNA. We confirmed the successful production and purity of the RNA/DNA hybrids by dot-blot assay and native-PAGE (*Figure 1—figure supplement 1f and g*). A 8-cycle difference between Tn5 transposome sample (Ct = 22.68) and the control sample (Ct = 30.40) was reproducibly observed (*Figure 1—figure supplement 1h*). As a positive control, we also annealed two complementary ssDNA strands of the same 150bp-CLuc sequence to produce dsDNA and observed that a Ct value of 18.08 for the dsDNA sample (*Figure 1—figure supplement 1h*). While it is unclear this difference in tagmentation efficiency obtained from short oligos can be applied to long oligos, this result clearly demonstrates that Tn5 has a direct tagmentation activity towards RNA/DNA hybrids.

Having demonstrated the tagmentation activity of Tn5 on RNA/DNA hybrids, we then thought about its potential applications. RNA/DNA duplexes can be found in many in vivo scenarios, including but not limited to R-loop and chromatin-bound lncRNAs (*Santos-Pereira and Aguilera, 2015*; *Li and Fu, 2019*). Under in vitro conditions, RNA/DNA hybrids are also key intermediates in various molecular biology and genomics experiments. For instance, RNA has to be first reverse transcribed into cDNA in a traditional RNA-seq experiment so as to construct a library for sequencing. Because traditional RNA-seq library construction involves many laborious and time-consuming steps, including mRNA purification, fragmentation, reverse transcription, second-strand synthesis, end-repair and adaptor ligation, we attempted to replace the process using the tagmentation activity towards RNA/DNA duplexes. With the help of TRACE-seq, these steps are replaced with a 'one-tube' protocol (*Figure 2a*), which uses total RNA as input material and involves just three seamless steps (reverse transcription, tagmentation and strand extension and PCR), without the need for a second strand synthesis step. We first conducted TRACE-seq with 200 ng total RNA as input and tested several enzymes and conditions (*Supplementary file 1*); we observed very high correlation in gene-expression levels among three replicates, indicating TRACE-seq is highly reproducible (*Figure 2b*). To test the robustness of TRACE-seq, we performed the experiments with 20 ng and 2 ng total RNA. TRACE-seq results are again highly reproducible among replicates (*Figure 2—figure*

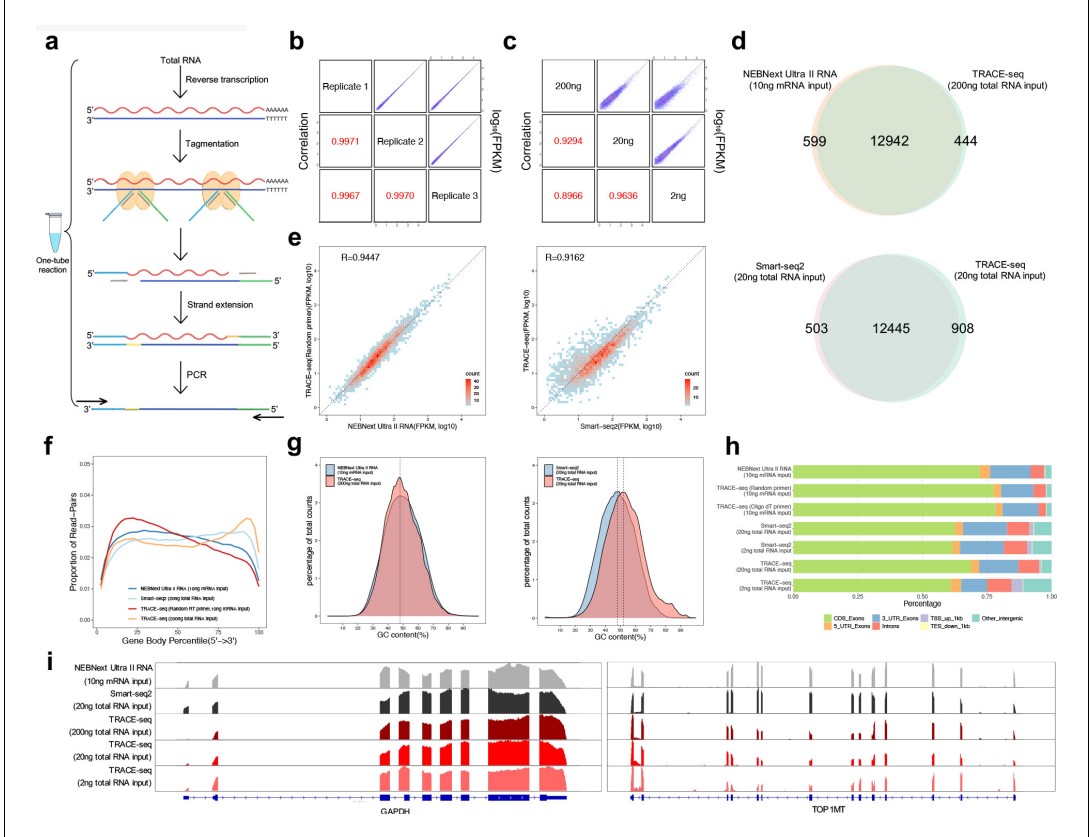

**Figure 2.** Workflow and evaluation of TRACE-seq. (a) Workflow of TRACE-seq. (b) Gene expression, measured by three technical replicates of TRACE-seq with 200 ng total RNA as input, are shown as scatter plots in the upper right half. Pearson's product-moment correlations are displayed in the lower left half. (c) Gene expression, measured by TRACE-seq using 200 ng, 20 ng and 2 ng total RNA as input, are shown as scatter plots in the upper right half. Pearson's product-moment correlations are displayed in the lower left half. (d) Venn diagrams of gene numbers detected by TRACE-seq with 200 ng total RNA as input and NEBNext Ultra II RNA kit with 200 ng mRNA as input (top) and by TRACE-seq with 20 ng total RNA as input and Smart-seq2 with 20 ng mRNA as input (below). (e) Scatterplots showing a set of housekeeping gene expression values for TRACE-seq and NEBNext Ultra II RNA kit with 10 ng mRNA as input (left), and for TRACE-seq with 10 ng mRNA as input and Smart-seq2 with 20 ng total RNA as input (right). Pearson's product-moment correlation is displayed in the upper left corner. (f) Comparison of read coverage over gene body for NEBNext Ultra II RNA kit, Smart-seq2 and TRACE-seq with different amount of RNA as input. The read coverage over gene body is displayed along with gene body percentile from 5' to 3' end. (g) Distribution of GC content of all mapped reads from TRACE-seq library with 200 ng total RNA as input and NEBNext Ultra II RNA library with 10 ng mRNA as input (left) or Smart-seq2 library with 20 ng total RNA as input (right). The vertical dashed lines indicate 48% (left) and 48% and 51% respectively (right). (h) Comparison of the distribution of reads across known genome features for NEBNext Ultra II RNA kit, Smart-seq2 and TRACE-seq with different amount of RNA as input. (i) IGV tracks showing the coverage of two representative transcripts (GAPDH and TOP1MT). The data come from NEBNext Ultra II RNA kit, Smart-seq2 and three sets of TRACE-seq with different amount of total RNA as input.

The online version of this article includes the following source data and figure supplement(s) for figure 2:

**Source data 1.** Distribution of reads across known genome features for NEBNext Ultra II RNA kit, Smart-seq2 and TRACE-seq with different amount of RNA as input.

**Figure supplement 1.** Quality assessment of TRACE-seq.

supplement 1a,b). More importantly, gene expression levels measured using different amount of starting materials remain consistent with each other (*Figure 2c*).

We then compared the library quality between TRACE-seq and NEBNext Ultra II RNA library prep kit, a commonly used kit for RNA-seq library construction. In addition, we conducted a comparison to Smart-seq2, which is a similar method in its use of oligo(dT) primed cDNA synthesis and Tn5 tagmentation. The HEK293T RNA used in these libraries was all from the same batch of cells. We found that TRACE-seq libraries exhibited similar percentage of reads mapped to annotated transcripts, rRNA contamination and gene numbers to NEBNext data when mRNA was used as input, but a higher rRNA contamination when total RNA was used as input (~9%, *Supplementary file 2*). In

addition, a similar percentage of rRNA contamination was also observed in Smart-seq2 libraries (*Supplementary file 3*). Most of the genes detected by TRACE-seq overlap with that of NEBNext and Smart-seq2 (*Figure 2d*). In addition, TRACE-seq showed comparable performance in terms of gene expression measurement, using either a set of housekeeping genes (*Figure 2e*) or all genes (*Figure 2—figure supplement 1c*). The insert size of TRACE-seq library was moderately shorter (*Figure 2—figure supplement 1d*); in the meanwhile, TRACE-seq shows a higher coefficient of variation of gene coverage (0.54–0.70 vs 0.42–0.44, *Figure 2—figure supplement 1e*). We further found that TRACE-seq showed a slight tendency to 3′ end of the gene body (*Figure 2f*, *Figure 2—figure supplement 1f*). When transcripts were grouped according to annotated lengths, we found comparable gene body coverage for transcripts shorter than 1 kb among TRACE-seq, NEBNext kit and Smart-seq2 libraries. For transcripts with length between 1 and 4 kb, a slight 3′ end bias was observed in TRACE-seq library, while for transcripts longer than 4 kb, the central regions of transcripts were less covered by both TRACE-seq and Smart-seq2. We also performed TRACE-seq by using rRNA depletion together with random-primed cDNA synthesis. While this solved the 3′ end bias, a 5′ end bias appeared, which is a common phenomenon when using random primers during reverse transcription (*Figure 2—figure supplement 1f*). In spite of the gene body coverage bias, GC content (*Figure 2g*) and library complexity (*Figure 2—figure supplement 1g*) are unnoticeably affected. In addition, the gene expression measurement (*Figure 2e*) is also unaffected here because of the use of RNA with high quality (RIN: 9.5, *Figure 2—figure supplement 1h*); yet, cautions might be taken when the quality of RNA is compromised. Further inspection of reads distribution of TRACE-seq over genome features revealed similar pattern for that of NEBNext and Smart-seq2 (*Figure 2h*). Coverages of some representative transcripts are shown in *Figure 2i*.

One of the most important goals of RNA-seq is to detect differentially expressed genes among different samples. Having assessed the library quality of TRACE-seq, we next compared the performance of TRACE-seq in detecting differentially expressed genes between undifferentiated and differentiated mESCs to NEBNext Ultra II RNA library prep kit. As shown in *Figure 3a*, TRACE-seq successfully detected 4577 differentially expressed genes (3264 up-regulated genes and 1313 down-regulated genes), while NEBNext detected 4452 differentially expressed genes (3157 up-regulated genes and 1295 down-regulated genes). The overlapping gene number is 4,071, showing very high consistency between methods (*Figure 3b*). Besides, the fold change of the 4071 overlapping genes is highly correlated between the two methods (R > 0.99, *Figure 3c*). Therefore, TRACE-seq shows excellent performance in differential gene expression analysis.

Previous studies found that Tn5 exhibits a slight insertion bias on dsDNA substrates (*Goryshin et al., 1998*; *Green et al., 2012*; *Lodge et al., 1988*). To further investigate whether potential bias exists for TRACE-seq, we thus characterized sites of Tn5-catalyzed adaptor insertion by calculating nucleotide composition of the first 30 bp of each sequence read per library. Similar to dsDNA substrates, we also observed an apparent insertion signature on RNA/DNA hybrids (*Figure 2—figure supplement 1i*). Nevertheless, per-position information contents were extremely low, suggesting such insertion bias is less likely to affect gene body coverage (*Figure 2—figure supplement 1j*). Overall, in spite of gene body coverage bias, TRACE-seq allows construction of high complexity RNA libraries and demonstrates similar performance as traditional RNA library preparation methods in terms of detected genes (97% and 93% overlapped with NEBNext and Smart-seq2 library respectively), gene expression measurement (R > 0.90) and differential expression analysis (R > 0.99), but outcompetes the traditional methods in terms of speed, convenience and cost.

## Discussion

Based on substrate diversity and the conserved catalytic domain of the retroviral integrase superfamily including the Tn5 transposase, we envision in this study that Tn5 may be able to directly tagment RNA/DNA hybrid duplexes, in addition to its canonical dsDNA substrates. Having validated such in vitro tagmentation activity, we developed TRACE-seq, which enables one-tube, low-input and low-cost library construction for RNA-seq experiments and demonstrates excellent performance in DE analysis. Compared to conventional RNA-seq methods, TRACE-seq does not need to pre-extract mRNA and synthesize a second DNA chain after mRNA reverse transcription. Therefore, TRACE-seq bypasses laborious and time-consuming processes, is compatible with low input, and reduces reagent cost (*Supplementary file 4*). During the preparation of this paper, an independent study

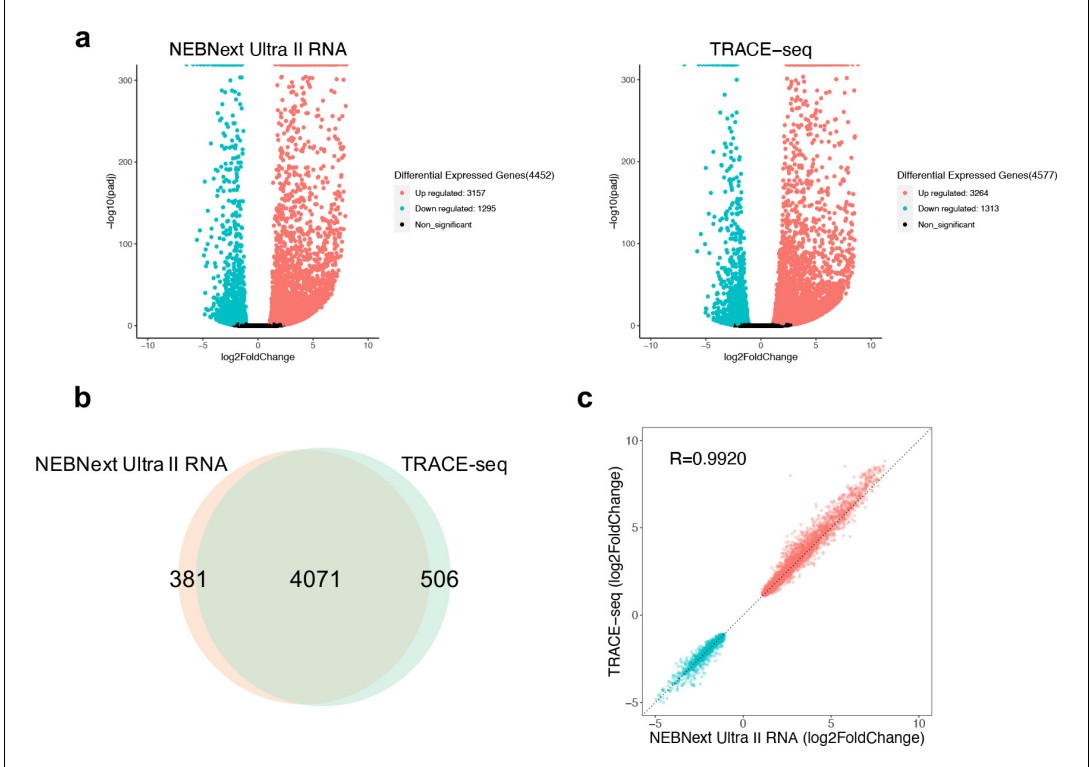

**Figure 3.** Performance of TRACE-seq in differential expression analysis. (**a**) Volcano plot showing differential expressed genes between undifferentiated and differentiated mESCs detected by NEBNext Ultra II RNA kit and TRACE-seq. Significantly up-regulated and down-regulated expressed genes (padj <0.05, |log2FoldChage| > 1) are highlighted in red and blue, respectively. (**b**) Venn diagram of differentially expressed gene numbers detected by TRACE-seq and NEBNext Ultra II RNA kit. (**c**) Correlation between the fold change of the 4071 differentially expressed genes that overlap between NEBNext Ultra II RNA and TRACE-seq library.

also reported the similar finding and developed a RNA-seq method named SHERRY (*Di et al., 2020*). The major conclusions are very consistent between the two studies.

Despite its unique advantages, there is room to further improve TRACE-seq. For instance, the libraries generated by TRACE-seq in its current form are not strand-specific, which is a significant drawback for RNA-Seq experiments. Yet, TRACE-seq should be able to be converted to 5' RNA-seq or 3' RNA-seq, which can directionally preserve the 5' or 3' end information of transcripts (*Cole et al., 2018*; *Pallares et al., 2020*). In addition, TRACE-seq could be used in multiplex profiling when utilizing Tn5 transposase containing barcoded adaptors (*Cusanovich et al., 2015*; *Zhu et al., 2019*). Besides, if home-made Tn5 can be used (as have done in *Picelli et al., 2014a*; *Kia et al., 2017*; *Kaya-Okur et al., 2019*), the costs will be further cut down. The in vitro tagmentation efficiency of Tn5 on RNA/DNA hybrids could also be further improved. We have shown that the addition of PEG200 substantially enhanced the tagmentation efficiency of hybrids. It is also tempting to speculate that hyperactive mutants towards RNA/DNA hybrids could also be obtained through screening and protein engineering, as wild-type Tn5 transposase has been engineered to obtain hyperactive forms (*Goryshin and Reznikoff, 1998*; *Wiegand and Reznikoff, 1992*; *Weinreich et al., 1994*; *Zhou and Reznikoff, 1997*). Such hyperactive mutants are expected to have immediate utility in single-cell RNA-seq experiments, for instance. Moreover, Tn5 transposition in vivo has been harnessed to profile chromatin accessibility in ATAC-seq (*Buenrostro et al., 2013*); it remains to be seen whether or not an equivalent version may exist to enable in vivo detection of R-loop, chromatin bound long non-coding RNA and epitranscriptome analysis (*Santos-Pereira and Aguilera, 2015*; *Li and Fu, 2019*; *Li et al., 2016*; *Song et al., 2020*). To summarize, TRACE-seq manifests a 'cryptic' activity of the Tn5 transposase as a powerful tool, which may have broad biomedical applications in the future.

## Materials and methods

### Cell culture

HEK293T cells (RRID:CVCL_1926) used in this study were daily maintained in DMEM medium (GIBCO) supplemented with 10% FBS (GIBCO) and 1% penicillin/streptomycin (GIBCO) at 37°C with 5% $CO_2$. We have confirmed no mycoplasma contamination using TransDetect PCR Mycoplasma Detection Kit (TransGen).

### Nucleic acids isolation

Total RNA was extracted from cells with TRIzol (Invitrogen), according to the manufacturer's instructions. The resulting total RNA was treated with DNase I (NEB) to avoid genomic DNA contamination. Phenol/chloroform extraction and ethanol precipitation were then performed to purify and concentrate total RNA. For mRNA isolation, two successive rounds of poly(A)+ selection were performed using oligo(dT)$_{25}$ dynabeads (Invitrogen). Genomic DNA (gDNA) from HEK293T cells was purified using genomic DNA purification kit (Qiagen) according to the manufacturer's instructions.

### RNA integrity number (RIN) assessment

Assessment of RNA integrity was performed with RNA 6000 Pico kits (Agilent Technologies). HEK293T total RNA sample was analyzed by Agilent 2100 Bioanalyzer (Agilent Technologies), and RIN was calculated using the supplied 2100 software.

### Preparation of RNA/DNA hybrids

A model mRNA (IRF9,~1000 nt) was in vitro transcribed from PCR products and purified by urea-PAGE. The model mRNA, HEK293T total RNA and mRNA were reverse transcribed into RNA/DNA hybrids by SuperScript IV reverse transcriptase (Invitrogen), according to the manufacturer's protocol, with several modifications: 1) Instead of oligo d(T)$_{20}$ primer, oligo d(T)$_{23}$VN primer (NEB) was annealed to template RNA; 2) Instead of SS IV buffer, SS III buffer supplemented with 7.5% PEG8000 was added to the reaction mixture; 3) The reaction was incubated at 55°C for 2 hr. To test the presence of RNA/DNA hyrids, IRF9 RT products were treated by DNase I (NEB) and RNase H (NEB) respectively according to the manufacturer's instructions followed by urea-PAGE analysis.

To generate the 150 bp RNA/DNA hybrid, the CLuc DNA template used for in vitro transcription (5'-TTAGCTTCACAGGAAGTTGGAACTGTGTTTGGTGGATCAGGTTCGTAAGGACAGTCCTGGCAAT TGAACAGTGGCGCAGTAGACTAATGCAACGGCAAGAATTAAGGTCTTCATGGTGGCGGA TCCGAGCTCGGTACCAAGCTTGGGTCTC-3') was first amplified by PCR from CLuc Control plasmid (NEB) with forward primer (5'-TAATACGACTCACTATAGGG-3') and reverse primer (5'-TTAGC TTCACAGGAAGTTGG-3'). RNA was produced from CLuc DNA template using in vitro transcription reaction by MAXIscript T7 Transcription Kit (Invitrogen). The resulting 150 nt RNA was treated by DNase I (NEB) and further purified by 6% urea-PAGE. Annealing between the purified in-vitro transcribed RNA and the synthesized complementary CLuc ssDNA sequence was conducted under two different conditions. 400 ng and 240 ng RNA was annealed with 200 ng DNA in group 1 and 2 respectively in the annealing buffer (50 mM Tris–HCl pH 7.6, 250 mM NaCl and 5 mM EDTA). The samples were first incubated for 5 min at 94°C and then cooled down slowly (1°C per minute) to room temperature. The annealed products were then purified using 2.2X Agencourt RNAClean XP beads (Beckman Coulter).

### Preparation of annealed dsDNA

To generate a 150 bp dsDNA, two complementary ssDNA strands were chemically synthesized and purified by 10% urea-PAGE. The resulting forward and reverse ssDNA strands were annealed under two different conditions. 400 ng and 240 ng forward ssDNA was annealed with 200 ng reverse ssDNA in group 1 and 2 respectively in the same annealing buffer as above. The annealing and purification procedure were performed as above.

### Characterization of 150 bp RNA/DNA hybrids by PAGE and Dot blot

The presence of RNA/DNA hybrids in the CLuc annealed products were confirmed by dot blot assay. Nitrocellulose membrane (Amersham Hybond-N+, GE) was marked and spotted with mRNA RT

products, 150 bp CLuc annealed products and 150 bp dsDNA (negative control). The membrane was air dried for 5 min before UV-crosslink (2X auto-crosslink, 1800 UV Stratalinker, STRATAGENE). After crosslinking, the membrane was blocked by 5% non-fat milk in 1X TBST at room temperature for 1 hr. Then the membrane was incubated with anti-hybrid S9.6 antibody (Kerafast, #ENH001, RRID:AB_2687463, 1:2000 dilution in 5% milk) for 1 hr at room temperature, followed by washing three times with 1X TBST. Lastly, the membrane was incubated with HRP linked anti-mouse secondary antibody (CWBiotech, RRID:AB_2736997) for 1 hr at room temperature. Signals were detected with ECL Plus Chemiluminescent reagent (Thermo Pierce).

The purity of RNA/DNA hybrids in the 150 bp CLuc annealed products were confirmed by 10% native-PAGE. Samples were loaded in an equal volume of native loading buffer (30% (v/v) glycerol, 80 mM HEPES-KOH (pH 7.9), 100 mM KCl, 2 mM magnesium acetate) and electrophoresed in 0.5 X TBE buffer at 180 V for 1.5 hr.

gDNA contamination detection qPCR experiments were performed to assess potential gDNA contamination. After DNase treatment, RNA samples were subjected to reverse transcription (RT). Two other groups (without RT enzyme and without RNA) were set as negative controls. These groups were subjected to qPCR with three pairs of primers respectively, using the method described above. The qPCR primers were designed within exons near the 3' end of three representative housekeeping genes:

GAPDH-qFWD: 5'-GCATCCTGGGCTACACTGAG-3';
GAPDH-qRVS: 5'-AAAGTGGTCGTTGAGGGCAA-3';
ACTB-qFWD: 5'-AGTCATTCCAAATATGAGATGCGTT-3';
ACTB-qRVS: 5'-TGCTATCACCTCCCCTGTGT-3';
CYC1-qFWD: 5'-CACCATAAAGCGGCACAAGT-3';
CYC1-qRVS: 5'-CAGGATGGCAAGCAGACACT-3'.

## Tn5 in vitro tagmentation on RNA/DNA hybrids

Partial double-stranded adaptor A and B were obtained by separately annealing 10 µM Tn5ME-A oligonucleotides (5'-TCGTCGGCAGCGTCAGATGTGTATAAGAGACAG-3') and Tn5ME-B oligonucleotides (5'-GTCTCGTGGGCTCGGAGATGTGTATAAGAGACAG-3') with equal amounts of mosaic-end oligonucleotides (5'-CTGTCTCTTATACACATCT-3') in annealing buffer (10 mM Tris–HCl pH 7.5, 10 mM NaCl). Samples were incubated for 5 min at 94°C and then cooled down slowly (1°C per minute) to 10°C. Assembly of Tn5 (TruePrep Tagment Enzyme, Vazyme, #S601-01) with equimolar mixture of annealed Adaptor A and B was performed according to the manufacturer's protocol (Vazyme). The resulting assembled Tn5 was stored at −20°C until use.

Tagmentation reaction was set up by adding RNA/DNA hybrids (RT products or CLuc annealed products) or gDNA, 12 ng/µl assembled Tn5 and 1 U/µl SUPERase-In RNase Inhibitor (Invitrogen) to the reaction buffer containing 10 mM Tris-HCl, pH = 7.5, 5 mM $MgCl_2$, 8% PEG8000% and 5% PEG200. The reaction was performed at 55°C for 30 min, and then SDS was added to a final concentration of 0.04% and Tn5 was inactivated for 5 min at room temperature.

## Assays of tagmentation activity of Tn5 on RNA/DNA hybrids

The concentrations of RNA/DNA hybrids and dsDNA were first determined by PicoGreen quantification kit (Invitrogen). For testing tagmentation activity of Tn5 on RNA/DNA hybrids, reactions were carried out as above, with mRNA derived RT products or CLuc annealed products as substrate. The tagmentation products were then purified using 1.8X Agencourt RNAClean XP beads (Beckman Coulter) to remove Tn5 and excess free adaptors and eluted in 6 µl nuclease-free water. The size distribution of RNA/DNA hybrids after tagmentation was assessed by a Fragment Analyzer Automated CE System with DNF-474 High Sensitivity NGS Fragment Analysis Kit (AATI).

For testing tagmentation activity of Tn5 on RNA/DNA hybrids by quantitative polymerase chain reaction (qPCR), tagmentation products purified as above (100X-diluted) was firstly strand-extended with 0.32 U/µl Bst 3.0 DNA Polymerase (NEB) and 1X AceQ Universal SYBR qPCR Master Mix (Vazyme) at 72°C for 15 min, and then Bst 3.0 Polymerase was inactivated at 95°C for 5 min. After adding 0.2 µM qPCR primers (5'-AATGATACGGCGACCACCGAGATCTACACTCGTCGGCAGCGTC-3'; 5'-CAAGCAGAAGACGGCATACGAGATGTCTCGTGGGCTCGG-3'), qPCR was performed in a LightCycler (Roche) with a 5 min pre-incubation at 95°C followed by 40 cycles of 10 s at 95°C and

40 s at 60℃. For testing the effect of different buffers on tagmentation activity of Tn5 on RNA/DNA hybrids, buffers used were as follows: 1) Tagment buffer L (Vazyme); 2) Buffer with 8% PEG8000 (10 mM Tris-HCl at pH 7.5, 5 mM $MgCl_2$ and 8% PEG8000); 3) Buffer with 10% DMF (10 mM Tris-HCl at pH 7.5, 5 mM $MgCl_2$ and 10% DMF); 4) Buffer with 5% PEG200% and 8% PEG8000 (10 mM Tris-HCl at pH 7.5, 5 mM $MgCl_2$, 5% PEG200% and 8% PEG8000).

## Sanger sequencing

The PCR products following RNA/DNA hybrid tagmentation and strand extension were ligated to a blunt-end cloning vector using pEASY-Blunt Zero Cloning Kit (TransGen), followed by chemical transformation. Then, several single colonies were picked and sequenced with the forward primers of T7 and T3 promoters.

## TRACE-seq library preparation and sequencing

For TRACE-seq library preparation, all reactions were performed in one tube. Reverse transcription and tagmentation reactions were carried out as above. Strand extension reaction was performed by directly adding 0.32 U/μl Bst 3.0 DNA Polymerase and 1X NEBNext Q5 Hot Start HiFi PCR Master Mix (NEB) to tagmentation products and incubating at 72℃ for 10 min, followed by Bst 3.0 DNA Polymerase inactivation at 95℃ for 5 min. Next, 0.2 μM indexed primers were added to perform enrichment PCR as follows: 30 s at 98℃, and then n cycles of 10 s at 98℃, 75 s at 65℃, followed by the last 10 min extension at 65℃. The PCR cycles 'n' depends on the amount of purified total RNA input (200 ng, n = 11; 20 ng, n = 14; 2 ng, n = 18). After enrichment, the library was purified twice using 1X Agencourt AMPure XP beads (Beckman Coulter) and eluted in 10 μl nuclease-free water. The concentration of resulting libraries was determined by Qubit 2.0 fluorometer with the Qubit dsDNA HS Assay kit (Invitrogen) and the size distribution of libraries was assessed by a Fragment Analyzer Automated CE System with DNF-474 High Sensitivity NGS Fragment Analysis Kit (AATI). Finally, libraries were sequenced on the Illumina Hiseq X10 platform which generated 2 × 150 bp of paired-end raw reads.

## NEBNext and Smart-seq2 library preparation

NEBNext Ultra II RNA libraries were constructed using NEBNext Ultra II RNA Library Prep Kit for Illumina (NEB, #E7770S) according to the manufacturer's instructions. Smart-seq2 libraries were performed according to the previously published protocol (*Picelli et al., 2014b*).

## Data analysis

Raw reads from sequencing were firstly subjected to Trim Galore (v0.6.4_dev, RRID:SCR_011847) (http://www.bioinformatics.babraham.ac.uk/projects/ trim_galore/) for quality control and adaptor trimming. The minimal threshold of quality was 20, and the minimal length of reads to remain was set as 20 nt. In terms of differential gene expression analysis, we down-sampled reads per library to 60 million. Otherwise, we down-sampled reads per library to 30 million. Then reads were mapped to human genome (hg19) and transcriptome using STAR (v2.7.1a, RRID:SCR_015899) (*Dobin et al., 2013*), and the transcriptome was prepared based on the Refseq annotation of human (hg19) downloaded from the table browser of UCSC database. rRNA contamination was determined through directly mapping to the dataset of human rRNA sequence downloaded from NCBI (NR_003286.2, NR_003287.2, NR_003285.2, and X71802.1) by bowtie2 (v2.2.9, RRID:SCR_005476) (*Langmead and Salzberg, 2012*). Performances related to the processing of sam/bam file were done with the help of Samtools (v1.9, RRID:SCR_002105) (*Li et al., 2009*). The FPKM value for annotated genes was calculated by cuffnorm (v2.2.1, RRID:SCR_014597) (*Trapnell et al., 2010*), and genes with FPKM >0.5 were considered to be expressed. Log-transformed FPKM values of housekeeping genes (*Supplementary file 5*, list from *Eisenberg and Levanon, 2013*) were plotted when comparison of gene expression measurement among TRACE-seq2, NEBNext and Smart-seq2 libraries. Gene body coverage and nucleotide composition for each position of the first 30 bases of each sequence read per library were calculated by QoRTs (v1.1.6, RRID:SCR_018665) (*Hartley and Mullikin, 2015*). Reads distribution and GC content distribution of mapped reads were calculated by RseQC (v2.6.4, RRID:SCR_005275) (*Wang et al., 2012*), and median coefficient of variation of gene coverage over the 1000 most highly expressed transcripts per library and insert size of library were calculated by

Picard Tools (v2.20.6, RRID:SCR_006525) (http://broadinstitute.github.io/picard/). Library complexity was calculated by Preseq (v2.0.0, RRID:SCR_018664) (*Daley and Smith, 2013*). The sequence conservations of Tn5 insertion sites on RNA/DNA hybrids were analyzed by WebLogo (v2.8.2, RRID: SCR_010236) (https://weblogo.berkeley.edu/). Reads Coverage was visualized using the IGV genome browser (v2.4.16, RRID:SCR_011793) (*Robinson et al., 2011*). Differential gene expression analysis was conducted using DESeq2 (v1.26.0, RRID:SCR_015687) (*Love et al., 2014*) with gene count data generated by HTSeq (v 0.11.2, RRID:SCR_005514) (*Anders et al., 2015*). And all corresponding graphs were plotted using R scripts by RStudio (v1.2.5033, RRID:SCR_000432) (https://rstudio.com/).

## Acknowledgements

The authors would like to thank Drs. Peng Du and Zhifang Zhang, Ms. June Liu for assistance with experiments and Mr. Dongsheng Bai for discussions. We thank National Center for Protein Sciences at Peking University in Beijing, China, for assistance with quantification of RNA/DNA hybrids, evaluation of tagmentation efficiency and library size distribution. Part of the analysis was performed on the High Performance Computing Platform of the Center for Life Science (Peking University). This work was supported by the National Natural Science Foundation of China (nos. 31861143026, 91740112 and 21825701 to CY) and Ministry of Science and Technology of China (nos. 2019YFA0110900 and 2019YFA0802201 to CY)

## Additional information

### Funding

| Funder | Grant reference number | Author |
| --- | --- | --- |
| National Natural Science Foundation of China | 31861143026 | Chengqi Yi |
| National Natural Science Foundation of China | 91740112 | Chengqi Yi |
| Ministry of Science and Technology of the People's Republic of China | 2019YFA0110900 | Chengqi Yi |
| Ministry of Science and Technology of the People's Republic of China | 2019YFA0802201 | Chengqi Yi |
| National Natural Science Foundation of China | 21825701 | Chengqi Yi |

The funders had no role in study design, data collection and interpretation, or the decision to submit the work for publication.

### Author contributions

Bo Lu, Conceptualization, Validation, Investigation, Visualization, Methodology, Writing - original draft; Liting Dong, Danyang Yi, Validation, Investigation, Visualization, Methodology, Writing - original draft; Meiling Zhang, Validation, Investigation, Methodology, Writing - review and editing; Chenxu Zhu, Conceptualization; Xiaoyu Li, Investigation; Chengqi Yi, Conceptualization, Supervision, Funding acquisition, Project administration, Writing - review and editing

### Author ORCIDs

Bo Lu (ID) https://orcid.org/0000-0002-5852-0477
Liting Dong (ID) http://orcid.org/0000-0001-8396-374X
Chenxu Zhu (ID) http://orcid.org/0000-0003-4216-6562
Chengqi Yi (ID) https://orcid.org/0000-0003-2540-9729

Decision letter and Author response
Decision letter https://doi.org/10.7554/eLife.54919.sa1
Author response https://doi.org/10.7554/eLife.54919.sa2

## Additional files

### Supplementary files

• Supplementary file 1. Quality control of the sequencing results using different enzymes for strand extension and PCR in TRACE-seq library construction procedure.

• Supplementary file 2. Quality control of the sequencing results using NEBNext kit and TRACE-seq.

• Supplementary file 3. Quality control of the sequencing results using Smart-seq2 and TRACE-seq.

• Supplementary file 4. Costs of RNA-seq constructed by NEBNext Ultra II RNA kit, TRACE-seq and Smart-seq2.

• Supplementary file 5. List of housekeeping genes.

• Transparent reporting form

### Data availability

High-throughput sequence data has been deposited in GEO under accession code GSE143422.

The following dataset was generated:

| Author(s) | Year | Dataset title | Dataset URL | Database and Identifier |
|---|---|---|---|---|
| Lu B, Dong L, Yi D, Zhang M, Yi C | 2020 | Transposase assisted tagmentation of RNA/DNA hybrid duplexes | https://www.ncbi.nlm.nih.gov/geo/query/acc.cgi?acc=GSE143422 | NCBI Gene Expression Omnibus, GSE143422 |

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

## Appendix 1

**Appendix 1—key resources table**

| Reagent type (species) or resource | Designation | Source or reference | Identifiers | Additional information |
|---|---|---|---|---|
| Cell line (*Homo-sapiens*) | HEK293T | American Type Culture Collection | Cat#: CRL-11268, RRID:CVCL_1926 | |
| Antibody | Mouse anti-DNA-RNA Hybrid [S9.6] Antibody | Kerafast | Cat#: ENH001, RRID:AB_2687463 | 1:2000 |
| Antibody | Antibody Anti-mouse-IgG-HRP | CWBiotech | Cat#: CW0102, RRID:AB_2736997 | 1:3000 |
| Recombinant DNA reagent | CLuc Control Template | NEB | Cat#: E2060S | |
| Sequence-based reagent | CLuc Control_F | This paper | PCR primers | TAATACGACTCACTATAGGG |
| Sequence-based reagent | CLuc Control_R | This paper | PCR primers | TTAGCTTCACAG-GAAGTTGG |
| Sequence-based reagent | GAPDH-qFWD | This paper | PCR primers | GCATCCTGGGCTA-CACTGAG |
| Sequence-based reagent | GAPDH-qRVS | This paper | PCR primers | AAAGTGGTCG TTGAGGGCAA |
| Sequence-based reagent | ACTB-qFWD | This paper | PCR primers | AGTCATTCCAAA TATGAGATGCGTT |
| Sequence-based reagent | ACTB-qRVS | This paper | PCR primers | TGCTATCACCTCCCC TGTGT |
| Sequence-based reagent | CYC1-qFWD | This paper | PCR primers | CACCATAAAGCGG-CACAAGT |
| Sequence-based reagent | CYC1-qRVS | This paper | PCR primers | CAGGATGGCAAGCA-GACACT |
| Sequence-based reagent | Tn5ME-A | doi: 10.1186/gb-2010-11-12-r119 | Transposon adaptor oligonucleotides | TCGTCGGCAGCGTC AGATGTGTATAAGA-GACAG |
| Sequence-based reagent | Tn5ME-B | doi: 10.1186/gb-2010-11-12-r119 | Transposon adaptor oligonucleotides | GTCTCGTGGGCTCGG AGATGTGTATAAGA-GACAG |
| Sequence-based reagent | Tn5MErev | doi: 10.1186/gb-2010-11-12-r119 | Transposon adaptor oligonucleotides | CTGTCTCTTATACACA TCT |
| Sequence-based reagent | Tn5_qFWD | This paper | PCR primers | AATGATACGGCGAC-CAC CGAGATCTACACTCG T CGGCAGCGTC |
| Sequence-based reagent | Tn5_qRVS | This paper | PCR primers | CAAGCAGAAGACGGC ATACGAGATGTCTC GTGGGCTCGG |
| Sequence-based reagent | TSO | doi:10.1038/nprot.2014.006 | Template switch primer | AAGCAGTGGTATCAA CGCAGAGTACATrGrG +G |
| Sequence-based reagent | ISPCR oligo | doi:10.1038/nprot.2014.006 | PCR primers | AAGCAGTGGTATCA ACGCAGAGT |

*Appendix 1—key resources table continued*

| Reagent type (species) or resource | Designation | Source or reference | Identifiers | Additional information |
|---|---|---|---|---|
| Sequence-based reagent | oligo dT(23)VN primer | NEB | Cat#: S1327S | |
| Sequence-based reagent | Random primer mix | NEB | Cat#: S1330S | |
| Sequence-based reagent | N501 primer | Illumina | PCR primers for sequencing | |
| Sequence-based reagent | N701-N712 primers | Illumina | PCR primers for sequencing | |
| Commercial assay or kit | TRIzol | Invitrogen | Cat#: 15596018 | |
| Commercial assay or kit | Blood & Cell Culture DNA Midi Kit | Qiagen | Cat#: 13343 | |
| Commercial assay or kit | MAXIscript T7 Transcription Kit | Invitrogen | Cat#: AM1314M | |
| Commercial assay or kit | SUPERase-In RNase Inhibitor | Invitrogen | Cat#: AM2696 | |
| Commercial assay or kit | Quant-iT PicoGreen dsDNA Assay Kit | Invitrogen | Cat#: P11496 | |
| Commercial assay or kit | AceQ Universal SYBR qPCR Master Mix | Vazyme | Cat#: Q511-02 | |
| Commercial assay or kit | pEASY-Blunt Zero Cloning Kit | TransGen | Cat#: CB501-01 | |
| Commercial assay or kit | NEBNext Q5 Hot Start HiFi PCR Master Mix | NEB | Cat#: M0544 | |
| Commercial assay or kit | Agencourt AMPure XP beads | Beckman Coulter | Cat#: A63882 | |
| Commercial assay or kit | RNAClean XP beads | Beckman Coulter | Cat#: A63987 | |
| Commercial assay or kit | Qubit dsDNA HS Assay kit | Invitrogen | Cat#: Q33230 | |
| Commercial assay or kit | DNF-474 High Sensitivity NGS Fragment Analysis Kit | Agilent | Cat#: DNF-473-1000 | |
| Commercial assay or kit | NEBNext Ultra II RNA Library Prep Kit for Illumina | NEB | Cat#: E7770S | |
| Commercial assay or kit | Dynabeads Oligo (dT)25 | Invitrogen | Cat#: 61005 | |
| Commercial assay or kit | KAPA HiFi HotStart ReadyMix | KAPA Biosystems | Cat#: KK2601 | |
| Commercial assay or kit | *TransDetect PCR Mycoplasma Detection Kit* | TransGen | Cat#: FM311-01 | |
| Commercial assay or kit | RNA 6000 Pico kits (Agilent Technologies | Agilent | Cat#: 5067-1513 | |
| Peptide, recombinant protein | DNase I | NEB | Cat#: M0303S | |

*Appendix 1—key resources table continued*

| Reagent type (species) or resource | Designation | Source or reference | Identifiers | Additional information |
|---|---|---|---|---|
| Peptide, recombinant protein | SuperScript IV reverse transcriptase | Invitrogen | Cat#: 12594100 | |
| Peptide, recombinant protein | SuperScript II reverse transcriptase | Invitrogen | Cat#: 18064022 | |
| Peptide, recombinant protein | RNase H | NEB | Cat#: M0297 | |
| Peptide, recombinant protein | TruePrep Tagment Enzyme | Vazyme | Cat#: S601-01 | |
| Peptide, recombinant protein | Bst 3.0 DNA Polymerase | NEB | Cat#: M0374S | |
| Chemical compound, drug | PEG200 | Sigma | Cat#: 88440 | |
| Chemical compound, drug | PEG8000 | Sigma | Cat#: 89510 | |
| Software, algorithm | Trim Galore | http://www.bioinformatics.babraham.ac.uk/projects/trim_galore/ | RRID:SCR_011847 | v0.6.4_dev |
| Software, algorithm | STAR | PMID:23104886 | RRID:SCR_015899 | v2.7.1a |
| Software, algorithm | bowtie2 | https://doi.org/10.1038/nmeth.1923 | RRID:SCR_005476 | v2.2.9 |
| Software, algorithm | Samtools | http://samtools.sourceforge.net/ | RRID:SCR_002105 | v1.9 |
| Software, algorithm | cuffnorm | PMID:20436464 | RRID:SCR_014597 | v2.2.1 |
| Software, algorithm | QoRTs | https://doi.org/10.1186/s12859-015-0670-5 | RRID:SCR_018665 | v1.1.6 |
| Software, algorithm | RseQC | PMID:22743226 | RRID:SCR_005275 | v2.6.4 |
| Software, algorithm | Picard Tools | http://broadinstitute.github.io/picard/ | RRID:SCR_006525 | v2.20.6 |
| Software, algorithm | Preseq | PMID:23435259 | RRID:SCR_018664 | v2.0.0 |
| Software, algorithm | RStudio | https://rstudio.com/ | RRID:SCR_000432 | 1.2.5033 |
| Software, algorithm | Integrative Genomics Viewer | http://software.broadinstitute.org/software/igv/ | RRID:SCR_011793 | v2.4.16 |

