## [Decision Letter]

**Acceptance summary:**

Your revised manuscript addresses the reviewers' prior concerns. We anticipate your new TRACE-Seq method will be of interest to readers as an efficient, lower cost alternative to traditional library construction methods for RNA-Seq.

**Decision letter after peer review:**

Thank you for sending your article entitled "Transposase assisted tagmentation of RNA/DNA hybrid duplexes" for peer review at *eLife*. Your article is being evaluated by three peer reviewers, and the evaluation is being overseen by a Reviewing Editor and Kevin Struhl as the Senior Editor.

As noted in our prior communications about the competing study that has now been published in PNAS, please be sure to mention that published study appropriately in your revised manuscript and to cite it in the main text.

Also, with regard to the name of your method, I agree with reviewer #1 that ATRAC-Seq does not make sense as an abbreviation and may be confused with ATAC-Seq, and recommend naming it something else.

Reviewer #1:

In Lu et al. the authors describe a strategy for producing RNA-seq libraries by the direct tagmentation of RNA-DNA hybrids. This method "ATRAC-seq" is very similar to "SHERRY" recently published in PNAS ("RNA sequencing by direct tagmentation of RNA/DNA hybrids") relying on what appears to be transposition activity of the Tn5 transposase into RNA/DNA hybrids. Overall the paper does a good job characterizing the RNA-seq libraries; however, like the PNAS publication, the authors do not have any experiments that explicitly test the RNA/DNA transposition efficiency. Neither the published work or the manuscript presented here take into account the various efficiencies of RT enzymes / mixes to produce dsDNA, varying based on the RNase H efficiency. It is worth noting that this is irrelevant for producing a simplified assay – it does not matter if the Tn5 is inserting into dsDNA product after the first strand synthesis or to the RNA/DNA hybrids, as both produce a simplified workflow for producing RNA-seq libraries. The issue is that any publication claiming this phenomenon without direct evidence in a controlled setting could result in misguided assumptions to the field. A properly controlled test would eliminate the RT component and directly assess hybrid constructs where no dsDNA is possible. It may be that the efficiency is high, which drives this result and not the dsDNA after RT; however, it needs to be directly demonstrated.

Other than the RNA-DNA transposition assumptions, the rest of the manuscript is a test of the RNA-seq libraries that were generated when compared to standard techniques, which are fairly standard and properly assessed.

Reviewer #2:

The manuscript "Transposase assisted tagmentation of RNA/DNA hybrid duplexes" by Lu et al. describes a new approach involving direct tagmentation of RNA/DNA heteroduplexes for a "one tube" mRNA-seq library preparation protocol called ATRAC-seq. Involving fewer steps, this workflow is allowing the generation of transcriptomics data with a seemingly similar quality as a conventional RNA-seq workflow and is reportedly faster and cheaper.

Indeed, as a novel approach, direct tagmentation of RNA/DNA hybrids looks very interesting and can potentially provide new grounds for improving a number of existing RNA-seq protocols allowing to bypass the second strand synthesis step.

The major concern, however, is the novelty of this work. A paper describing very similar results and a comparable transcriptomics approach have just been published recently as a peer-review article (Da et al., PNAS) and last November 2019 as a preprint. Importantly, some authors from the current Lu et al. work seem to be affiliated with the same departments as the co-authors on Da et al., namely the Tsinghua-Peking Center for Life Sciences and College of Chemistry and Molecular Engineering, Peking University. This might be considered as merely an unlucky coincidence, but the overall similarity of the two works is truly puzzling and thus suggests this might not be the case. It involves obvious parallels in the overall flow of the manuscript and its structure: 1) rationale for attempting the tagmentation of hybrids with Tn5; 2) experimental approach; 3) workflow for mRNA-seq benchmarking; 4) figure layouts look (i.e. Figure 1 in both works show protein domain structure similarity between RNAse H superfamily members). The actual RNA-seq method ATRAC-seq described by the authors is apparently identical to SHERRY from Da et al., with slight variations such as the enzyme (Superscript II vs Superscript IV; Bst2 vs Bst3) and tagmentation buffer composition (9% PEG8000 vs 8% PEG8000). In brief, one may think that a number of merely esthetical changes were introduced in the work of Lu et al. to make it appear distinct from Da et al., 2020. That being said, the work of Da et al. also provides more details and mechanistic insights concerning tagmentation of hybrids.

Finally, the benchmarking is rather meager, as at a minimum, differential gene expression should be included as well as other parameters as for example detailed in Levin et al., Nat Meth, 2010; Alpern et al., Genome Biology, 2019; Pallares et al., 2020.

Other comments:

• How does the 256-fold increase in number of amplifiable fragments after tagmentation with active Tn5 vs inactive with RNA/DNA hybrids compare when dsDNA is used as a substrate? In this regard, what the authors may have addressed is the basis of tagmentation efficiency of RNA/DNA hybrids and dsDNA. It would be interesting to know what drives the preference of the Tn5 for tagmenting one substrate over another.

• The RNA samples might be contaminated with gDNA, which will presumable serve as a better substrate for Tn5, did the authors check this possibility experimentally or by checking the resulting sequencing reads?

• What was the reason of using Bst polymerase? Have the authors compared this to the results obtained with the conventional tagmentation protocol involving PCR amplification as described in the protocol of Picceli et al., 2014? This also relates to the shallow benchmarking already mentioned above.

Reviewer #3:

General Assessment:

This manuscript presents a new method termed "ATRAC-seq," which uses Tn5 to fragment RNA/cDNA hybrids to streamline RNA-Seq library construction. This is an interesting advancement, though the standard methods are not that difficult or time-consuming (contrary to the authors' statements). For this method to be widely adopted, the authors would need to show more data about quality and address issues such as Tn5 sequence specificity and 3' coverage bias. Note that essentially the same method has been published on January 27, 2020 as "SHERRY" -- https://www.pnas.org/content/early/2020/01/24/1919800117.

Numbered summary of any substantive concerns.

1) One key problem with the manuscript is that the authors do not use a standard sample for which the expression values are known, so that the comparison with the NEBNext Ultra II RNA library prep kit is inconclusive. It's not possible to know whether there is "comparable performance" as written about Figure 2E without a known standard or another control. The authors should add a new series of experiments with standard samples, such as "the well-characterized reference RNA samples A (Universal Human Reference RNA) and B (Human Brain Reference RNA) from the MAQC consortium, adding spike-ins of synthetic RNA from the External RNA Control Consortium (ERCC)" as published by the SEQC/MAQC-III Consortium in Nature Biotechnology 32:903-914 (2014). In that paper, the authors compare to qRT-PCR data as well as RNA-Seq. Moreover, Figure 2E shows R=0.6970 between the NEB and ATRAC-seq libraries – that is not particularly good correlation.

2) In addition, it would be interesting to see a comparison to Smart-seq2, which is a similar method in its use of oligo(dT) primed cDNA synthesis and Tn5 tagmentation. This method is much closer to ATRAC-seq than the NEB kit.

3) The authors need to more explicitly address 3' end bias (as shown in Figure 2F), as it relates to sequence coverage of genes based on their length. Analysis could be presented as in Figure 1 of Ramsköld et al. Nature Biotechnology 30:777-782 (2012). The 3' end bias was also observed in Di et al. PNAS (Figure S11 and page 7). How will this affect expression level measurements and downstream analysis? One possible solution is to use rRNA depletion together with random-primed cDNA synthesis?

4) Other analyses that should be considered are evenness of coverage along a transcript (coefficient of variation) and the ability to identify differentially expressed genes (Figure 3B in Di et al.).

5) The authors should explain further the "per-position" analysis (Figure 1—figure supplement 1F) as it is not clear what is being shown or how it was calculated.

6) There are experimental and computational details missing from this manuscript. The authors should add the following:

a) Were the ATRAC-seq and NEB libraries prepared from the same RNA?

b) What was the RIN score of RNA used in each experiment?

c) How was the NEB library constructed? This is not mentioned in the Materials and methods section.

d) How is the annealing done for Tn5 oligos (concentration, time, temperature, buffers)?

e) What is the full name and catalog # for the Tn5 purchased from Vazyme?

f) How many reads are there for each library? Analyses should be done with the same number of raw reads per library by down-sampling.

g) Accession #'s for human rRNA should be listed.

[Editors' note: further revisions were suggested prior to acceptance, as described below.]

Thank you for submitting your article "Transposase assisted tagmentation of RNA/DNA hybrid duplexes" for consideration by *eLife*. Your article has been reviewed by three peer reviewers, and the evaluation has been overseen by a Reviewing Editor and Kevin Struhl as the Senior Editor. The following individuals involved in review of your submission have agreed to reveal their identity: Bart Deplancke (Reviewer #2).

The reviewers have discussed the reviews with one another and the Reviewing Editor has drafted this decision to help you prepare a revised submission.

Summary:

The revisions have addressed most of the concerns raised previously by the reviewers. Some additional revisions need to be carried out before the manuscript is acceptable for publication.

Revisions expected in follow-up work:

1) See comment by reviewer #1 regarding a missing positive control and comparisons of efficiency for DNA/RNA hybrids versus dsDNA.

2) A number of concerns made by reviewer #3 regarding the presentation in the manuscript. None of these concerns require additional experiments.

Reviewer #1:

I appreciate that the authors went to a good deal of work to test the synthetic constructs that they describe. They note Ct values of 24 (active Tn5), 28 (inactivated Tn5), and 29 (negative control); however, they neglect to include a positive control. I am surprised, as this would be an easy addition – annealing a ssDNA to the other ssDNA template as opposed to RNA. As it stands, the ct of 24 seems very late for transposed product. Comparing to dsDNA will give a sense of the difference in efficiency between DNA/RNA hybrids and dsDNA. The other edits are fine, this is the last component I believe needs to be addressed.

Reviewer #2:

The authors have adequately addressed our major concerns. No further comments.

Reviewer #3:

General Assessment:

The revised manuscript is much improved. It was good to see the addition of the Smart-seq2 and rRNA depletion with TRACE-seq experiments. It is understandable that the authors could not add an experiment with a standard reference sample or spike-ins due to the COVID-19 outbreak. There are still issues remaining with respect to analysis, presentation, and conclusions.

Numbered summary of any substantive concerns.

1) The authors' use of housekeeping genes to assess correlation in gene expression measurements between different methods is acceptable, but these issues should be addressed.

a) The use of a set of housekeeping genes should be clearly identified in the Results section and the figure legends

b) The actual names of the housekeeping genes used should be listed in a Supplementary table rather than "list from Eisenberg and Levanon, 2013)" as in the Materials and methods section.

c) They should also present the analysis with all the genes – noting that one example is shown in the authors' response to reviewers' comments.

2) In several places, the authors minimize the underperformance of their TRACE-seq method. In each place, the authors should modify the text and include the actual numbers for the readers in the text. Finally, the text should be modified from "are comparable", "demonstrates comparable performance", and "shows comparable performance" to something more measured that lists the advantages and disadvantages.

a) Coefficient of Variation is actually much worse (Figure 2—figure supplement 1D) not "slightly higher coefficient of variation".

b) 5' to 3' bias. This issue is not apparent here because of the use of high quality RNA (RIN 9.5), but with lower quality "real world" samples, the bias becomes more of an issue and the gene expression measurements will be affected. This should be noted in relation to the statement "In spite of the gene body coverage bias, the gene expression measurement (Figure 2E).… [is] unnoticeably affected."

c) rRNA-aligned reads is actually ~100x worse for 200ng total RNA than for 10ng mRNA. That is not "slightly higher but acceptable". It is probably acceptable, but that's a judgement for the reader to make.

d) Strandedness. The authors now do mention this, but this is actually a significant drawback for RNA-Seq experiments.

3) The authors' explanation of the "per-position" analysis (Figure 2—figure supplement 2I) as still not clear about what is being shown or how it was calculated.

4) The cost comparison between NEBnext and TRACE-seq is good, but the Smart-seq2 should be included and it is likely less expensive than either method ($10-15/library).

---

## [Author Response]

Reviewer #1:In Lu et al. the authors describe a strategy for producing RNA-seq libraries by the direct tagmentation of RNA-DNA hybrids. This method "ATRAC-seq" is very similar to "SHERRY" recently published in PNAS ("RNA sequencing by direct tagmentation of RNA/DNA hybrids") relying on what appears to be transposition activity of the Tn5 transposase into RNA/DNA hybrids. Overall the paper does a good job characterizing the RNA-seq libraries; however, like the PNAS publication, the authors do not have any experiments that explicitly test the RNA/DNA transposition efficiency. Neither the published work or the manuscript presented here take into account the various efficiencies of RT enzymes / mixes to produce dsDNA, varying based on the RNase H efficiency. It is worth noting that this is irrelevant for producing a simplified assay – it does not matter if the Tn5 is inserting into dsDNA product after the first strand synthesis or to the RNA/DNA hybrids, as both produce a simplified workflow for producing RNA-seq libraries. The issue is that any publication claiming this phenomenon without direct evidence in a controlled setting could result in misguided assumptions to the field. A properly controlled test would eliminate the RT component and directly assess hybrid constructs where no dsDNA is possible. It may be that the efficiency is high, which drives this result and not the dsDNA after RT; however, it needs to be directly demonstrated.

We thank this reviewer for the suggestion. To address this question, we directly tested Tn5 tagmentation activity on RNA/DNA hybrids produced independently of reverse transcription reaction. Between homopolymers (for instance, poly[rA:dT] or poly[rI:dC]) and base-diversified hybrids, we chose the latter since it is a better mimic of real substrates for RNA-seq experiments. Because the limit of ssDNA length produced by commercially available solid-state chemical synthesis is about 150 nt, we aimed to produce 150 bp RNA/DNA hybrids independently of reverse transcription. We first produced ssRNA model sequence (CLuc, 150 nt, GC%=51%) by in vitro transcription reaction from PCR products. Then we annealed the IVT ssRNA with a 150 nt synthesized complementary ssDNA to produce a RNA/DNA hybrid. We confirmed the successful production of annealed CLuc RNA/DNA hybrids by dot-blot assay and the purity of the resulting hybrids was further examined by native-PAGE (Figure 1—figure supplement 1E and 1F). We then subjected the prepared RNA/DNA hybrids to Tn5 transposome, heat-inactivated Tn5 transposome and a blank control (without Tn5) followed by qPCR quantification. We observed that cycle threshold (Ct) value of the Tn5 transposome sample (Ct=24.86) is about 4 cycles smaller than the heat inactivated Tn5 sample (Ct=28.89) or the control sample (Ct=29.25), indicating that Tn5 has direct tagmentation activity towards RNA/DNA hybrids (Figure 1—figure supplement 1G). We have incorporated all the above results into the revised manuscript (Figure 1—figure supplement 1E, 1F and 1G, Results section).

Other than the RNA-DNA transposition assumptions, the rest of the manuscript is a test of the RNA-seq libraries that were generated when compared to standard techniques, which are fairly standard and properly assessed.

We thank the reviewer for the positive comment.

Reviewer #2:The manuscript "Transposase assisted tagmentation of RNA/DNA hybrid duplexes" by Lu et al. describes a new approach involving direct tagmentation of RNA/DNA heteroduplexes for a "one tube" mRNA-seq library preparation protocol called ATRAC-seq. Involving fewer steps, this workflow is allowing the generation of transcriptomics data with a seemingly similar quality as a conventional RNA-seq workflow and is reportedly faster and cheaper.Indeed, as a novel approach, direct tagmentation of RNA/DNA hybrids looks very interesting and can potentially provide new grounds for improving a number of existing RNA-seq protocols allowing to bypass the second strand synthesis step.The major concern, however, is the novelty of this work. A paper describing very similar results and a comparable transcriptomics approach have just been published recently as a peer-review article (Da et al., PNAS) and last November 2019 as a preprint. Importantly, some authors from the current Lu et al. work seem to be affiliated with the same departments as the co-authors on Da et al., namely the Tsinghua-Peking Center for Life Sciences and College of Chemistry and Molecular Engineering, Peking University. This might be considered as merely an unlucky coincidence, but the overall similarity of the two works is truly puzzling and thus suggests this might not be the case. It involves obvious parallels in the overall flow of the manuscript and its structure: 1) rationale for attempting the tagmentation of hybrids with Tn5; 2) experimental approach; 3) workflow for mRNA-seq benchmarking; 4) figure layouts look (i.e. Figure 1 in both works show protein domain structure similarity between RNAse H superfamily members). The actual RNA-seq method ATRAC-seq described by the authors is apparently identical to SHERRY from Da et al., with slight variations such as the enzyme (Superscript II vs Superscript IV; Bst2 vs Bst3) and tagmentation buffer composition (9% PEG8000 vs 8% PEG8000). In brief, one may think that a number of merely esthetical changes were introduced in the work of Lu et al. to make it appear distinct from Da et al., 2020. That being said, the work of Da et al. also provides more details and mechanistic insights concerning tagmentation of hybrids.

We thank this reviewer for pointing out the competing study. We did not know about it when we designed our project; despite the fact that some the authors of PNAS are our colleagues, we did not talk about the projects throughout the entire study and the two studies are independently performed. When we were about to submit our work at the end of 2019, we did then notice their paper posted on bioRxiv. To expedite our study, we quickly prepared our manuscript and also indicated the competing study in our initial cover letter to *eLife*. Thus, since the initial submission of our work, we aimed to make all the information we have as transparent as possible to the editorial office and reviewers. Now that their work published in PNAS, we have properly cited the study in the Discussion section of our revised manuscript.

With regard to the similarity of figure layout and experimental details of our method (renamed to “TRACE-seq” per the request of reviewer #1) with SHERRY: we acknowledge that the two methods have similar concept and thus similar reagents that are key to the success of the tagmentation activity on RNA/DNA hybrids. This could be further magnified by the simplified library preparation procedure, which now contain greatly reduced steps comparing to traditional library preparation procedure. This is especially the case in terms of some key reagents. For example, PEG8000 is a known crowding agent that effectively increased the efficiency of tagmentation reaction (Picelli et al., 2014). Nevertheless, these independently developed conditions and chosen reagents demonstrate that Tn5-mediated tagmentation activity is reproducible and robust.

In order to provide more details and mechanistic insights concerning tagmentation of hybrids, we have performed multiple experiments during the revision. For instance, we have provided evidence that Tn5 has direct tagmentation activity towards RNA/DNA hybrids (Figure 1—figure supplement 1G). This is achieved by assessing Tn5-mediated tagmentation activity on RNA/DNA hybrids obtained without reverse transcriptase, which could produce dsDNA from RNA/DNA hybrids. We designed an experiment by eliminating the RT component and directly assess annealed RNA/DNA hybrid constructs where no dsDNA is possible. We confirmed the successful production and purity of annealed RNA/DNA hybrids by dot-blot assay and native-PAGE (Figure 1—figure supplement 1E and 1F), and observed activity of Tn5 on such RNA/DNA hybrids substrate. Secondly, during revision we discovered a new component that greatly improves Tn5 tagmentation efficiency towards RNA/DNA hybrids (Figure 1E). The addition of PEG200 in the tagmentation condition, which we believe enables the RNA/DNA hybrids to favor B-form conformation (Pramanik et al., 2011), makes hybrids a better substrate of Tn5 and thus significantly improves the tagmentation reaction and quality of libraries. With this key invention, the current correlation between NEB and TRACE-seq (R=0.9072, All genes) is much better than the data in the original manuscript. Third, we have performed additional differential gene expression analysis using differentiated and undifferentiated mESCs. All of these results are unique to this study and have been incorporated into the revised manuscript (see Figure 1E, Figure 1—figure supplement 1E,1F,1G and Figure 3 A-C, Results section).

Finally, the benchmarking is rather meager, as at a minimum, differential gene expression should be included as well as other parameters as for example detailed in Levin et al., Nat Meth, 2010; Alpern et al., Genome Biology, 2019; Pallares et al., 2020.

We thank the reviewer for the suggestion and pointing out these references. Because it is well known that there are many differentially expressed genes between undifferentiated and differentiated mESCs (Bhattacharya et al., 2004; Hailesellasse Sene et al., 2007; Palmqvist et al., 2005), during revision, we chose them as inputs for differential gene expression analysis using TRACE-seq procedure and the traditional NEB procedure, respectively.

As shown in Figure 3A, TRACE-seq successfully detected 4,577 differentially expressed genes (3,264 up-regulated genes and 1,313 down-regulated genes), while NEB detected 4,452 differentially expressed genes (3,157 up-regulated genes and 1,295 down-regulated genes). The overlapping gene number is 4,071 (Figure 3B), showing very high consistency between methods. Besides, the fold change of the 4,071 overlapping genes is highly correlated between methods (R> 0.99, Figure 3C). Therefore, TRACE-seq shows excellent performance in differential gene expression analysis.

We also assessed the performance of TRACE-seq in terms of library complexity and evenness of coverage as mentioned in previous work (Levin et al., 2010). TRACE-seq (using random RT primer) library has high complexity, which is highly consistent with NEB library. What’s more, TRACE-seq library complexity maintains at high level with a small input material (Figure 2—figure supplement 1G). As for evenness of coverage, compared to NEB and Smart-Seq2 libraries, TRACE-seq libraries show a slightly higher coefficient of variation of gene coverage (Figure 2—figure supplement 1D). These results have been incorporated into the revised manuscript (Figure 2—figure supplement 1D and 1F, Results section).

Other comments:• How does the 256-fold increase in number of amplifiable fragments after tagmentation with active Tn5 vs inactive with RNA/DNA hybrids compare when dsDNA is used as a substrate? In this regard, what the authors may have addressed is the basis of tagmentation efficiency of RNA/DNA hybrids and dsDNA. It would be interesting to know what drives the preference of the Tn5 for tagmenting one substrate over another.

We thank this reviewer for the question. To answer this question, we performed tagmentation experiment on equal amount of gDNA and mRNA RT products (quantified by PicoGreen assay) followed by qPCR experiment to quantify amplifiable fragments. We found that average Ct value (Ct=25.23) of the hybrids samples was about 4 cycles more than gDNA samples (Average Ct value=21.19), indicating the efficiency of Tn5 towards hybrids is about 1/16 of that of dsDNA.

It is known that natural RNA/DNA hybrids favor A-form conformation. Interestingly, in the presence of PEG200, hybrids were found to favor B-form conformation (Pramanik et al., 2011), which we expected to make the hybrids a better substrate of Tn5. Indeed, addition of PEG200 diminishes this difference by greatly improving the Tn5 tagmentation efficiency towards hybrids (Ct=21.94), while having no influence on dsDNA (Ct=21.15). This result indicates that the conformation of substrates certainly affects the preference of Tn5. In addition, this finding also significantly improves the tagmentation reaction and provides a condition that greatly improves the quality of libraries. These results have been incorporated into the revised manuscript (Figure 1E, Results section).

• The RNA samples might be contaminated with gDNA, which will presumable serve as a better substrate for Tn5, did the authors check this possibility experimentally or by checking the resulting sequencing reads?

We appreciate the reviewer’s thoroughness. During revision, we have performed qPCR experiments to assess potential gDNA contamination. After DNase treatment, RNA samples were subjected to reverse transcription (RT). Two other groups (without RT enzyme and without RNA) were set as negative controls. Then we utilized primer pairs within an exon of three represented housekeeping genes to perform qPCR for these groups, respectively. In this way, potential gDNA contamination in the RNA sample and the cDNA generated by RT can be amplified simultaneously.

We found that for all the three genes, the RT groups have small Ct numbers, while the two negative control groups have very large (>35) or even not detectable (N.D.) Ct numbers, similar with qPCR result of blank control (water) (Figure 1—figure supplement 1H). Therefore, it is the newly generated cDNA via RT that is detected by qPCR, and we found no sign of gDNA contamination.

Meanwhile, we have also performed bioinformatic analysis of the reads distributions. One would expect to observe many reads from intron if the library were prepared from potential gDNA contamination. On the contrary, as shown in Figure 2H, TRACE-seq libraries starting from purified mRNA showed high exon distribution rates (94.04% from Random primer, and 95.99% from Oligo dT primer) and low intron distribution rates (4.88% and 2.91%). We thus conclude that the library is constructed from RNA/DNA hybrids after RT, not potential gDNA contamination. These results have been incorporated into the revised manuscript (Figure 1—figure supplement 1H, Figure 2H, Results section).

• What was the reason of using Bst polymerase? Have the authors compared this to the results obtained with the conventional tagmentation protocol involving PCR amplification as described in the protocol of Picceli et al., 2014? This also relates to the shallow benchmarking already mentioned above.

Per you request, we have constructed libraries using Bst 3.0 DNA polymerase and SS IV reverse transcriptase respectively and analyzed the sequencing data, so as to compare the strand extension catalytic performance of the enzymes. We found that the Bst 3.0 DNA polymerase showed higher mapping ratio (Supplementary file 1); we thus think Bst 3.0 may be a better choice for strand extension in TRACE-seq.

We have also constructed libraries using Q5 DNA polymerase and KAPA DNA polymerase (used in the protocol of Picceli et al., 2014) respectively to compare their PCR amplification performance. The mapping ratio of Q5 DNA polymerase library was significantly higher than KAPA DNA polymerase library. The results showed that Q5 DNA polymerase performed better than KAPA DNA polymerase during amplification of TRACE-seq (Supplementary file 1). In addition, due to addition of both PEG200 and DMF to tagmentation reaction, the mapping ratios of these libraries were overall lower than that of libraries tagmented without DMF. Thus, we choose to only add PEG200 to conventional tagmentation buffer in the final recipe to achieve greatly improved tagmentation efficiency.

Further benchmarking has been conducted during revision, including differential gene expression analysis, assessment of library complexity and evenness of coverage, which are incorporated into the revised manuscript (Figure 3, Figure 2—figure supplement 1D and 1G, Results section).

Reviewer #3:General Assessment:This manuscript presents a new method termed "ATRAC-seq," which uses Tn5 to fragment RNA/cDNA hybrids to streamline RNA-Seq library construction. This is an interesting advancement, though the standard methods are not that difficult or time-consuming (contrary to the authors' statements). For this method to be widely adopted, the authors would need to show more data about quality and address issues such as Tn5 sequence specificity and 3' coverage bias. Note that essentially the same method has been published on January 27, 2020 as "SHERRY" -- https://www.pnas.org/content/early/2020/01/24/1919800117.Numbered summary of any substantive concerns.1) One key problem with the manuscript is that the authors do not use a standard sample for which the expression values are known, so that the comparison with the NEBNext Ultra II RNA library prep kit is inconclusive. It's not possible to know whether there is "comparable performance" as written about Figure 2E without a known standard or another control. The authors should add a new series of experiments with standard samples, such as "the well-characterized reference RNA samples A (Universal Human Reference RNA) and B (Human Brain Reference RNA) from the MAQC consortium, adding spike-ins of synthetic RNA from the External RNA Control Consortium (ERCC)" as published by the SEQC/MAQC-III Consortium in Nature Biotechnology 32:903-914 (2014). In that paper, the authors compare to qRT-PCR data as well as RNA-Seq. Moreover, Figure 2E shows R=0.6970 between the NEB and ATRAC-seq libraries – that is not particularly good correlation.

We agree with the reviewer that a more standard sample would give us more accurate information about the performance of TRACE-seq. Unfortunately, due to the outbreak of COVID-19, neither Universal Human Reference RNA, Human Brain Reference RNA nor ERCC was available at the moment. Instead of external spike-in RNA, internal control genes were also frequently used to normalize gene expression data (de Kok et al., 2005). Thus, we chose a series of endogenous control genes, i.e. housekeeping genes (list from Trends Genet 29, 569-574(2013)), to compare the performance of TRACE-seq and NEB kit in terms of gene expression measurement, and found a high correlation between NEB and TRACE-seq for endogenous control genes (R=0.9447). Thus, we believe TRACE-seq demonstrates comparable performance to the NEB kit. These results have been incorporated into the revised manuscript (Figure 2E, Results section).

In the meanwhile, we have significantly optimized the tagmentation condition during revision. Now, as shown below, the correlation between NEB and TRACE-seq (R=0.9072, All genes) is much better than the data in the original manuscript. This is due to the addition of PEG200 in the tagmentation condition, which we believe enables the RNA/DNA hybrids to favor B-form conformation and thus a better substrate of Tn5 (Pramanik et al., 2011). This finding significantly improves the tagmentation efficiency and provides a condition that greatly improves the quality of libraries. These results have been incorporated into the revised manuscript (Figure 1E, Result section).

2) In addition, it would be interesting to see a comparison to Smart-seq2, which is a similar method in its use of oligo(dT) primed cDNA synthesis and Tn5 tagmentation. This method is much closer to ATRAC-seq than the NEB kit.

We thank the reviewer for the suggestion. During revision, we have performed additional RNA-seq library construction by both TRACE-seq and Smart-seq2, and then compared their performance in terms of gene number, gene expression analysis, gene body coverage, evenness of coverage, library complexity, etc. in the revised manuscript. Most of the genes (~93%) detected by TRACE-seq overlap with that of Smart-seq2, with slightly more genes detected by TRACE-seq. In addition, TRACE-seq showed comparable performance to Smart-seq2 in terms of gene expression measurement. When the gene body coverage across all transcripts is concerned, TRACE-seq showed a slightly more 3’ end bias than Smart-seq2, consistent with the observation that TRACE-seq shows a slightly higher coefficient of variation of gene coverage. Nevertheless, TRACE-seq library complexity is similarly high when compared to NEB and Smart-seq2 library.

3) The authors need to more explicitly address 3' end bias (as shown in Figure 2F), as it relates to sequence coverage of genes based on their length. Analysis could be presented as in Figure 1 of Ramsköld et al. Nature Biotechnology 30:777-782 (2012). The 3' end bias was also observed in Di et al. PNAS (Figure S11 and page 7). How will this affect expression level measurements and downstream analysis? One possible solution is to use rRNA depletion together with random-primed cDNA synthesis?

Per the request of the reviewer, transcripts were grouped according to annotated lengths and the gene body coverage was analyzed separately. As shown in Figure 2 —figure supplement 1F, the gene body coverage was comparable among TRACE-seq, NEB kit and Smart-seq2 libraries for transcripts shorter than 1kb. For transcripts with length between 1kb and 4kb, a 3’ end bias was observed in TRACE-seq libraries (using oligo dT RT primer); for transcripts with length between 4kb and 15kb, the central regions of transcripts were less covered by TRACE-seq (using oligo dT RT primer), which is also a known phenomenon in Smart-seq2 libraries (Xiao et al., 2018). We also performed TRACE-seq by using rRNA depletion together with random-primed cDNA synthesis. While this indeed solved the 3’ end bias of TRACE-seq, we appear to observe a 5’ end bias. Nevertheless, this library shows a higher correlation with NEB kit (R=0.9447). In spite of these gene body coverage bias, the gene expression measurement and differential gene expression analysis is almost unaffected, as shown in Figure 2E and Figure 3.

4) Other analyses that should be considered are evenness of coverage along a transcript (coefficient of variation) and the ability to identify differentially expressed genes (Figure 3B in Di et al.).

We thank the reviewer for the suggestion. We have calculated the median coefficient of variation of coverage over the 1000 most highly expressed transcripts by Picard Tools in the revised manuscript. Compared to NEB and Smart-seq2 libraries, TRACE-seq libraries show a slightly high coefficient of variation of coverage (Figure 2 —figure supplement 1D).

In addition, comparison of the performance in terms of differential gene expression analysis between NEB and TRACE-seq libraries have been conducted in the revised manuscript. Because it is well known that there are many differentially expressed genes between undifferentiated and differentiated mESCs (Bhattacharya et al., 2004; Hailesellasse Sene et al., 2007; Palmqvist et al., 2005), during revision we performed DE analysis using TRACE-seq procedure and the traditional NEB procedure, respectively.

As shown in Figure 3A, TRACE-seq successfully detected 4,577 differentially expressed genes (3,264 up-regulated genes and 1,313 down-regulated genes), while NEB detected 4,452 differentially expressed genes (3,157 up-regulated and 1,295 down-regulated genes). The overlapping gene number is 4,071 (Figure 3B), showing very high consistency between methods. Besides, the fold change of the 4,071 overlapping genes is highly correlated between methods (R> 0.99, Figure 3C). Therefore, TRACE-seq shows excellent performance in differential gene expression analysis.

5) The authors should explain further the "per-position" analysis (Figure 1—figure supplement 1F) as it is not clear what is being shown or how it was calculated.

We apologize for not making this clear in the original main text. In the revised manuscript, we calculated nucleotide composition of the first 30 bases of each sequencing read per library by QoRTs to characterize the potential insertion bias of Tn5 (Figure 2—figure supplement 1H, I). The detailed description can be found in the Results section.

6) There are experimental and computational details missing from this manuscript. The authors should add the following:a) Were the ATRAC-seq and NEB libraries prepared from the same RNA?

Yes, we extracted RNA from cultured HEK293T cells and then used it in parallel to construct NEB, TRACE-seq and Smart-seq2 library. The newly updated results are shown in Figure 2 and Figure 2—figure supplement 1.

b) What was the RIN score of RNA used in each experiment?

We have performed experiment to assess the RIN score. The HEK293T RNA samples used in experiments are from the same batch. The RIN score of the batch is high as shown in Figure 2—figure supplement 1E, so the RNA integrity is assured.

c) How was the NEB library constructed? This is not mentioned in the Materials and methods section.

The NEB library was constructed according to the manufacturer’s instructions (NEB #E7770S). We have added several sentences in the revised manuscript (see the Materials and methods).

d) How is the annealing done for Tn5 oligos (concentration, time, temperature, buffers)?

10μM synthetic Tn5ME-A adapter or 10μM Tn5ME-B adapter were mixed with equal amount of Tn5ME-Rev oligos in annealing buffer (10 mM Tris–HCl pH 7.5, 10 mM NaCl). The two samples were both incubated in a PCR block starting at 95°C for 5min followed by decreasing gradient of 1°C per minute until 10°C. We have added several sentences in our revised manuscript (see the Materials and methods).

e) What is the full name and catalog # for the Tn5 purchased from Vazyme?

The full name of Vazyme Tn5 is “TruePrep Tagment Enzyme” and the catalog number is #S601-01 (see the Materials and methods).

f) How many reads are there for each library? Analyses should be done with the same number of raw reads per library by down-sampling.

Indeed, our analyses were done with the same number of reads per library. In terms of differential gene expression analysis, we down-sampled reads per library to 60 million. Otherwise, we down-sampled reads per library to 30 million.

g) Accession #'s for human rRNA should be listed.

We apologize for the missing information. The accession numbers for human rRNA we used are NR_003286.2, NR_003287.2, NR_003285.2, and X71802.1.

[Editors' note: further revisions were suggested prior to acceptance, as described below.]

Reviewer #1:I appreciate that the authors went to a good deal of work to test the synthetic constructs that they describe. They note Ct values of 24 (active Tn5), 28 (inactivated Tn5), and 29 (negative control); however, they neglect to include a positive control. I am surprised, as this would be an easy addition – annealing a ssDNA to the other ssDNA template as opposed to RNA. As it stands, the ct of 24 seems very late for transposed product. Comparing to dsDNA will give a sense of the difference in efficiency between DNA/RNA hybrids and dsDNA. The other edits are fine, this is the last component I believe needs to be addressed.

We thank this reviewer for the suggestion. Per your request, we bought commercially available 150nt ssDNA strands and produced dsDNA by annealing the two strands. The 150 bp-hybrid was prepared as previously mentioned. We then subjected the prepared RNA/DNA hybrids and dsDNA to Tn5 transposome, and a blank control (without Tn5) followed by qPCR quantification. We observed that cycle threshold (Ct) value of the hybrid sample with Tn5 transposome treatment (Ct=22.68) is about 8 cycles smaller than the control sample (Ct=30.4), indicating that Tn5 has direct tagmentation activity towards RNA/DNA hybrids. We also observed that cycle threshold (Ct) value of the dsDNA sample with Tn5 transposome treatment (Ct=18.08) is about 4 cycles smaller than the hybrid sample with Tn5 transposome treatment (Ct=22.68), indicating that the efficiency of Tn5 towards hybrids is about 1/16 of that of dsDNA (Figure 1—figure supplement 1G). It should be noted that we previously conducted qPCR assay with templates diluted 1:600, which might be over-diluted and led to the previously observed Ct value (24.86). Now we are conducting the assay with templates diluted 1:100. We have incorporated all the above results into the revised manuscript (Figure 1—figure supplement 1G and 1H, Results section).

Reviewer #3:General Assessment:The revised manuscript is much improved. It was good to see the addition of the Smart-seq2 and rRNA depletion with TRACE-seq experiments. It is understandable that the authors could not add an experiment with a standard reference sample or spike-ins due to the COVID-19 outbreak. There are still issues remaining with respect to analysis, presentation, and conclusions.Numbered summary of any substantive concerns.1) The authors' use of housekeeping genes to assess correlation in gene expression measurements between different methods is acceptable, but these issues should be addressed.a) The use of a set of housekeeping genes should be clearly identified in the Results section and the figure legends

We have included the words “a set of housekeeping genes” in the Results section and the figure legends in the revised manuscript.

b) The actual names of the housekeeping genes used should be listed in a Supplementary table rather than "list from Eisenberg and Levanon, 2013)" as in the Materials and methods section.

We have listed the names of the housekeeping genes in the Supplementary file 6.

c) They should also present the analysis with all the genes – noting that one example is shown in the authors' response to reviewers' comments.

We have presented the scatterplots with all genes as Figure 2—figure supplement 1C in the revised manuscript.

2) In several places, the authors minimize the underperformance of their TRACE-seq method. In each place, the authors should modify the text and include the actual numbers for the readers in the text. Finally, the text should be modified from "are comparable", "demonstrates comparable performance", and "shows comparable performance" to something more measured that lists the advantages and disadvantages.

We have modified the corresponding text to avoid the vague descriptions in the revised manuscript.

a) Coefficient of Variation is actually much worse (Figure 2—figure supplement 1D) not "slightly higher coefficient of variation".

We have modified the text and included the actual numbers of coefficient of variation in the revised manuscript.

b) 5' to 3' bias. This issue is not apparent here because of the use of high quality RNA (RIN 9.5), but with lower quality "real world" samples, the bias becomes more of an issue and the gene expression measurements will be affected. This should be noted in relation to the statement "In spite of the gene body coverage bias, the gene expression measurement (Figure 2E).… [is] unnoticeably affected.".

We have modified the text and reminded the readers to pay attention to the quality of RNA in the revised manuscript.

c) rRNA-aligned reads is actually ~100x worse for 200ng total RNA than for 10ng mRNA. That is not "slightly higher but acceptable". It is probably acceptable, but that's a judgement for the reader to make.

We have modified the text and included the actual percentage of rRNA contamination in the revised manuscript.

d) Strandedness. The authors now do mention this, but this is actually a significant drawback for RNA-Seq experiments.

We have modified the text to explicitly point out this drawback in the current version of TRACE-seq. We also discussed a potential approach to preserve the strand information.

3) The authors' explanation of the "per-position" analysis (Figure 2—figure supplement 2I) as still not clear about what is being shown or how it was calculated.

We apologize for the missing information. We have included more detailed information in the figure legend of Figure 2—figure supplement 1J in the revised manuscript.

4) The cost comparison between NEBnext and TRACE-seq is good, but the Smart-seq2 should be included and it is likely less expensive than either method ($10-15/library).

Firstly, we apologize for a mistake in the cost calculation of TRACE-seq in our previous manuscript. The cost of Tn5 per reaction of TRACE-seq should be $5.43 instead of $27.16. We previously divided the total price of Tn5 by the wrong number of rxn. The actual total cost of TRACE-seq should be $12.68 per reaction. Secondly, we found that the cost of Smart-seq2 in our lab should be $29.12, based on the commercially available reagents in China. This price is close to the cost ($35) previously reported (Supplementary table 5, Picelli S, Björklund Å K, Faridani O R, et al. Smart-seq2 for sensitive full-length transcriptome profiling in single cells[J]. Nature methods, 2013, 10(11): 1096-1098.). Overall, TRACE-seq is more cost-effective than NEBNext kit and Smart-seq2. We have included the cost of Smart-seq2 in the revised Supplementary file 4.